# Physical Activity among Italian Adolescents: Association with Life Satisfaction, Self-Rated Health and Peer Relationships

**DOI:** 10.3390/ijerph19084799

**Published:** 2022-04-15

**Authors:** Daniela Pierannunzio, Angela Spinelli, Paola Berchialla, Alberto Borraccino, Lorena Charrier, Paola Dalmasso, Giacomo Lazzeri, Alessio Vieno, Silvia Ciardullo, Paola Nardone

**Affiliations:** 1National Centre for Disease Prevention and Health Promotion, Italian National Institute of Health, Viale Regina Elena 299, 00161 Rome, Italy; daniela.pierannunzio@iss.it (D.P.); angela.spinelli@iss.it (A.S.); paola.nardone@iss.it (P.N.); 2Department of Clinical and Biological Sciences, University of Torino, Regione Gonzole 43, 10043 Orbassano, Italy; paola.berchialla@unito.it; 3Department of Public Health and Pediatrics, University of Torino, Via Santena 5 bis, 10126 Turin, Italy; alberto.borraccino@unito.it (A.B.); lorena.charrier@unito.it (L.C.); paola.dalmasso@unito.it (P.D.); 4Department of Molecular and Developmental Medicine, University of Siena, 53100 Siena, Italy; giacomo.lazzeri@unisi.it; 5Department of Developmental and Social Psychology, University of Padova, 35131 Padua, Italy; alessio.vieno@unipd.it

**Keywords:** physical activity, psychosocial wellbeing, adolescents, HBSC

## Abstract

The aim of this study was to describe physical activity, both moderate-to-vigorous physical activity (MVPA) and vigorous physical activity (VPA), in a large nationally representative sample of Italian adolescents, aged 11, 13 and 15, and to evaluate if Italian adolescents were in line with the 2020 WHO recommendations. In order to assess the possible impact of physical activity on adolescent psychological and social wellbeing, the associations between MVPA and VPA with life satisfaction, self-rated health and peer relationships were also explored. Data from the 2018 Italian Health Behaviour in School-aged Children (HBSC) survey on 58,976 adolescents were analysed. Logistic regression analysis was used to investigate the association between physical activity and self-rated health, life satisfaction and peer relationships. The association of the Family Affluence Scale with self-rated health, life satisfaction and peer relationships was also studied. The results showed that the majority of Italian adolescents did not meet current physical activity WHO guidelines on MPVA. The prevalence of adolescents claiming to be involved “at least four or more times per week” in VPA was 29%. Overall, our findings highlighted a positive association between MVPA and VPA and life satisfaction, self-rated health and peer support. Data from this study underlined the need to encourage physical activity, especially among older adolescents and girls, who claimed lower levels of MPVA and VPA.

## 1. Introduction

The improvement in physical activity and the reduction in sedentary behaviour among children, adolescents and adults have been addressed by the World Health Organization (WHO) and other international bodies during the last decades [1,2,3,4,5]. Possible implications for both the general health of the world population and the prevalence of major non-communicable diseases (NCDs) (i.e., cardiovascular disease, diabetes, obesity and cancer) and other NCD risk factors, such as hypertension, coronary heart disease and overweight, are well documented [6]. Further, mental health, quality of life and wellbeing improvement can be positively affected by an adequate level of physical activity [7]. In 2010, the WHO produced a document, which examined the direct relationship between physical activity and health benefits, and provided evidence-based recommendations on frequency, duration and intensity of physical activity for health enhancement and prevention of NCDs (WHO 2010), as well as for the improvement of mental health, emotional wellbeing and social relationships [8]. An update of these guidelines was proposed by the WHO in 2020, using the same approach, based on the evidence of association between physical activity, sedentary behaviour and specific health outcomes, and presented new recommendations for children, adolescents, adults and older adults [9]. Additionally, Canadian 24-h Movement Guidelines for children, adolescents, adults and older adults were recently developed by the Canadian Society for Exercise Physiology, with the aim to address the right amounts of physical activity, sedentary behaviours, as well as sleep for a healthy 24 h [10].

The WHO has recently estimated, at the global level, that more than 25% of adults and 80% of adolescents are insufficiently physically active [11]. Concerning young people, recent research has studied the prevalence and trends of insufficient physical activity among 1.6 million students, aged 11–17 years, by country, region, and globally [12]. In general, the results showed that the majority of adolescents did not meet the current physical activity WHO guidelines; that is, at least an average of 60 daily minutes of moderate-to-vigorous-intensity physical activity (MVPA), such as brisk walking, dance, and cycling to school, and vigorous-intensity physical activity (VPA), such as running, soccer, swimming laps, as well as those that strengthen muscles and bones, should be incorporated at least three times per week [9].

During adolescence, health-related behaviours, both physical and social, are adopted, which may persist throughout adulthood. Adolescence is a period of life for adopting lifelong health-related behaviours and several changes, both biological and social, that typically take place during this phase, may persist throughout adulthood [13]. In particular, physical activity plays an essential role in the improvement of physical, psychological and social wellbeing, as well as the quality of life of adolescents, with important health benefits for both body and mind among adolescents [14,15,16,17]. However, especially among young people, the time spent in physical activity may be replaced by more sedentary habits. During the last few decades, the enhancement of technology has caused a decrease in physical activity and, at the same time, an increase in sedentary activities, such as screen-based entertainment (television and computer) and digital communication and mobile phones [18]. In contrast, some studies showed that sedentary behaviours have no significant role in causing young people to avoid regular physical activity [19,20].

A study on trends in MVPA across 32 countries from Europe and North America, using data from the Health Behaviour in School-aged Children (HBSC) survey, showed only a small increase in the percentage of adolescents aged 11, 13 and 15 years meeting the recommendations from 2002 to 2010 [21]. The recent international report on the 2018 HBSC survey showed that only 19% of European 11-, 13- and 15-year-old adolescents achieved the WHO recommendations on MVPA. Moreover, participation in VPA according to the WHO guidelines was 49% and 35% for boys and girls, respectively, for all age groups. A higher level of physical activity among boys than girls, and an increase in gender differences with age for both MVPA and VPA was reported. The decrease in physical activity with age was more evident among girls [22].

Some authors have reported the possible impact of physical activity on overall mental health as well. In particular, it might contribute to reducing adolescents’ anxiety, distress, depression, and it might also improve self-esteem, life satisfaction and perception of health [14,23,24,25]. Recent studies on the HBSC survey showed the association between physical activity with life satisfaction, in addition to other mental health indicators, such as the WHO-5 index, Mental Health Inventory and the HBSC Symptom Checklist, highlighting gender and age differences [26,27].

Furthermore, physical activity during adolescence may provide social benefits, enhancing cohesion among peers and improving their relationships [24]. For social health, physical activity may also be important for vulnerable young people. In particular, the inclusion of adolescents living with physical disability and chronic conditions, and those who are socioeconomically disadvantaged, in physical activity may reduce inequalities. Moreover, physical activity may also facilitate the integration of migrants and foreign young people [28].

Socioeconomic status may influence the psychological and social wellbeing of adolescents. Data from the international HBSC 2018 highlighted the association between socioeconomic status, measured by the Family Affluence Scale (FAS), and some mental (i.e., life satisfaction and perception of health) and social (peer relationships) outcomes. The 2018 HBSC report also pointed out the strong association of socioeconomic status with physical activity, showing that adolescents from less affluent families claimed to be less involved in physical activities, both MPVA and VPA [22].

The aim of this study was to describe physical activity, both MVPA and VPA, in a large nationally representative sample of Italian adolescents, aged 11, 13 and 15 years, and to evaluate whether Italian adolescents were in line with the 2020 WHO recommendations on MVPA and VPA. In order to evaluate the possible impact of physical activity on adolescent psychological and social wellbeing, the associations between MVPA and VPA with life satisfaction, self-rated health and peer relationships were also explored. The association of FAS with self-rated health, life satisfaction and peer relationships was also studied. A stratification by gender and age was also included in the analysis, so as to identity possible differential impacts of physical activity among boys and girls and between 11–13 and 15-year-old adolescents.

## 2. Materials and Methods

Data were collected as part of the 2018 HBSC study. HBSC is a WHO Collaborative Cross-National Survey of school students, which collects data every four years on wellbeing, social environments and health behaviours in early adolescents (aged 11, 13, and 15 years). The last 2018 HBSC survey includes data from 50 countries across Europe and North America, all adhering to a detailed international study protocol [29].

The Italian survey involved a regional representative sample of students that filled out a self-completed anonymous questionnaire on their dietary habits, physical activity, risk behaviours and wellbeing, and their relationship with their school, parents and peers as well as general information concerning their health and social background [30,31].

Target variables on physical activity were measured by the questions “Over the past 7 days, on how many days were you physically active for a total of at least 60 min per day?” (the so-called “moderate-to-vigorous physical activity (MVPA)” with possible answers: 0 = 0 days to 7 = seven days) and “Outside school hours: how often do you usually exercise in your free time so much that you get out of breath or sweat?” (the so-called “vigorous physical activity” (VPA) with possible answers: 0 = never, 1 = less than once a month, 2 = once a month, 3 = once a week, 4 = 2 to 3 times a week, 5 = 4 to 6 times a week and 6 = every day) so as to investigate compliance with the WHO physical activity guidelines [9].

In the logistic regressions models MPVA was categorized into “at least 4 days” vs. “less than 4 days”. “Less than 4 days” of MVPA for 60 min during last seven days was identified as “low levels of moderate-to-vigorous physical activity” [32]. For VPA the cut-off “at least 2 times a week” was used to examine adolescents’ sport activities in relation to “at least three days a week” recommended by the WHO for VPA [9].

FAS is used as a proxy of socio-economic status of the family [33]. This validated indicator is based on 6 items: number of cars in the family (0 = “none”, 1 = “one”, 2 = “two or more”), having one’s own bedroom (0 = “no”, 1 = “yes”), number of computers (0 = “none”, 1 = “one”, 2 = “two”, and 3 = “more than two”), having a dishwasher (0 = “no”, 1 = “yes”), number of bathrooms (0 = “none”, 1 = “one”, 2 = “two”, 3 = “more than two”), and frequency of family holidays (0 = “not at all”, 1 = “once”, 2 = “twice”, 3 = “more than twice”). After summing the scores, FAS is categorised into three levels: ‘low affluence’ (0–6), ‘moderate affluence’ (7–9) and ‘high affluence’ (>10).

As measure of general life satisfaction and indicator of wellbeing, the Cantril ladder was used in which the top “10” means the best possible life and “0” the worst [34,35]. Students were asked “In general, where on the ladder do you feel you stand at the moment?” and in this study responders were categorized into those with normal to high life satisfaction (6–10) and those with low life satisfaction (0–5).

Students were also asked to indicate their health status answering the question “Would you say your health is…?”. The four possible answers were dichotomized into “excellent/good” vs. “fair/poor” self-rated health [36].

An overall score for measuring peer support was calculated by adding four item scores on the questions “My friends really try to help me”, “I can count on my friends when things go wrong”, “I have friends with whom I can share my joys and sorrows”, “I can talk about my problems with my friends” (possible answers range from 1 = very strongly disagree to 7 = very strongly agree) and dividing by four; a score of at least 5.5 points was used to define high peer support [37,38].

Logistic regression models were used to explore the association between physical activity, both MVPA and VPA, FAS and adolescents’ life satisfaction, self-rated health, peer relationships. The likelihood was described by odds ratios (OR) with their 95% confidence intervals (CI). Given the gender and age differences in physical activities, life satisfaction, self-rated health and peer support, and logistic regression analyses were stratified by these variables. Missing data were excluded from the analysis. Stata software version 16.1 was used for all statistical analyses STATA v16.1 (StataCorp, College Station, TX, USA).

## 3. Results

The Italian HBSC 2018 survey included 64,929 students from 4183 selected classes. After data cleaning and applying the inclusion criteria, 58,976 students’ data were eligible for analysis. For the single variables, a further 0.2 to 3.6% were missing. Table 1 shows the sample’s main features. For each age group, the male and female ratio is 1:1 (male: 29,820 vs. female: 29,156).

The prevalence rates of MVPA and VPA by age and gender are shown in Table 1. For MPVA, 12.2% of boys and 6.8% of girls reported achieving 60 daily minutes of MPVA over the past seven days. MPVA decreased with age for both boys and girls and, overall, prevalence fell from 14.8% to 8.5% among boys and from 8.9% to 5.1% among girls between the ages 11–15. Furthermore, 6.5% of boys and 11.4% of girls aged 11, 13 and 15 were not ever involved in MVPA during the past seven days. The percentage increased dramatically from age 11 to 15, doubling for boys and tripling for girls.

The enrolled adolescents often reported to be physically active in their free time, outside school hours, at least two or three times a week. Similar to MPVA, at all ages, levels of VPA were generally higher among boys than girls; 74.4% of boys and 61.6% of girls reported participating in vigorous physical activity at least two or three times a week. VPA participation decreased with age among boys and girls; significant gender differences were observed in all age groups and the largest gender gap was found among 15-year-old adolescents (approximately 15 percentage points). As such, 14.1% of boys and 22.6% of girls aged 11, 13 and 15 claimed to participate in VPA once a month, less than once a month or never. The prevalence of physically inactivity among girls was higher than that found for boys.

As reported in Table 1, the majority of adolescents claimed high levels of life satisfaction. In general, boys and younger adolescents felt satisfied with their lives; overall, the prevalence rates were 90.4% and 85.8% for boys and girls, respectively. Life satisfaction decreased with age for both boys and girls, but with some gender differences. In detail, life satisfaction among boys was comparable between ages 11 and 13, and a slight decrease was observed from 13- to 15-year-old adolescents. Concerning girls, the decline by age was stronger in comparison with boys and, in addition, it was also noticed between ages 11 and 13.

Regarding the perception of health, the majority of Italian adolescents reported “good/excellent” health. In general, 15-year-old teenagers declared lower rates of “good/excellent” health than younger adolescents, and the decline in “good/excellent” health perception was more pronounced among girls. The percentages of “good/excellent” health were, at all ages, 92.0% for boys and 89.4% for girls. For boys, the prevalence ranged from 92.8% for 13-year-old adolescents to 90.0% for 15 year olds. As regards girls, a prevalence range from 93.3% to 84.3% was observed for 13 and 15 years, respectively.

A high prevalence of adolescents claimed high support from their peers; a slight decrease in peer support was observed for both boys and girls from 11 to 13 years old, while no differences were found between ages 13 and 15. Overall, girls were more likely to report high levels of peer support (girls, 71.6% vs. boys, 61.3%), and gender differences remained stable with age increase.

As shown in Table 1, the majority of adolescents had a medium FAS level (boys: 47.2%; girls: 47.6%) and about one-third had a low FAS level.

The results of logistic regression models are reported in Table 2. Concerning physical activities, the results showed an association between both MVPA and VPA with life satisfaction, self-rated health and peer support. The association between life satisfaction and MVPA was significant only among 13-year-old boys (OR 1.59, 95% CI 1.22–2.07). The positive effect of VPA on life satisfaction was found to be higher among older boys (13 years: OR = 1.38, 95% CI = 1.05–1.82; 15 years: OR = 1.64, 95% CI = 1.29–2.10). In contrast, the association between VPA and life satisfaction decreased for girls aged from 11 to 15 years (11 years: OR = 1.36, 95% CI = 1.05–1.76; 13 years: OR = 1.27, 95% CI = 1.03–1.57; 15 years: OR = 1.19, 95% CI = 0.94–1.43).

The results on adolescents’ health perception showed that the positive effect of MVPA was greater among younger boys (11 years: OR = 2.23, 95% CI = 1.62–3.07; 13 years: OR = 1.56, 95% CI = 1.20–2.03; 15 years: OR = 1.57, 95% CI = 1.21–2.05). Among girls, the strongest positive association between MVPA and health perception was found in 13 year olds (13 years: OR = 1.53, 95% CI = 1.14–2.06). Furthermore, the findings highlighted that a higher frequency of vigorous physical activity (VPA) was associated with a major level of positive health perception, especially in boys. In contrast to MPVA, a positive association between a higher level of VPA and good/excellent health perception was observed among older boys (13 years: OR = 1.75, 95% CI = 1.30–2.34; 15 years: OR = 2.48, 95% CI = 1.92–3.21). A positive association was also found among girls for all age groups. The results revealed that 15-year-old girls who were mainly involved in VPA in their free time were more likely to claim a high level of health perception; no differences were observed for 11- and 13-year-old girls.

A positive association between a higher level of MVPA and peer support was found for boys and girls and the results were comparable at all age groups, with the exception of 13-year-old teenagers. In detail, 13-year-old boys showed the strongest positive association between peer support and MVPA (13 years: OR = 1.52, 95% CI = 1.31–1.77) but the results were in contrast with those concerning 13-year-old girls, which did no show the same association. The association of VPA with peer support among boys consisted of a major positive effect among older boys. Comparable results were found for 11- and 13-year-old girls, but no effect on peer support was found for 15-year-old girls.

The results showed that socioeconomic status (measured by FAS) was associated with psychological and social wellbeing in adolescents: a medium–high FAS was positively associated with higher life satisfaction, good/excellent health perception and better relationships with peers.

## 4. Discussion

The results showed that the majority of Italian adolescents did not meet current physical activity WHO guidelines [9]. The MVPA level was only found to be adequate (at least an average of 60 min per day) for about 10% of adolescents. In order to investigate the compliance with WHO guidelines on VPA (at least 3 days a week of VPA), the sum of the answers for “2 to 3 times a week or more” was used to estimate the prevalence of Italian adolescents meeting the WHO recommendations. The findings underlined a higher prevalence of VPA than MPVA; around 68% of Italian adolescents reported a level of VPA that was in agreement with the WHO recommendations. The prevalence of VPA among adolescents decreased to 29%, if the answer “2 to 3 times a week” was excluded (adding only the answers “4 to 6 times a week” and “every day”).

The comparison of the Italian HBSC 2018 results with the international HBSC 2018 showed a lower prevalence of MPVA among Italian adolescents (Italian, 10% vs. international, 19%) [18]. Similar to MVPA, the results on VPA “at least 4 or more times per week” were lower among Italian adolescents than those from other countries (Italian, 29% vs. international, 42%).

The data from 298 school-based surveys, from 146 countries, territories, and areas, including 1.6 million students aged 11–17 years, showed that the majority of adolescents did not conform to current WHO recommendations on MPVA for health [12].

The Italian HBSC results showed significant age and gender differences for physical activity among adolescents and these findings were comparable with the international HBSC. In particular, physical activity participation decreased with age, and girls tended to do less MVPA and VPA than boys. For MPVA, the gender difference decreased with age in the Italian HBSC, in contrast with the international results. On the other hand, the gender differences increased with age for VPA and the results were comparable to the international HBSC 2018 [22]. Guthold et al. (2020) noted the same marked gender differences, confirming that participation in physical activity is lower among girls. In addition, an improvement in physical activity prevalence was pointed out for boys between 2001 and 2016, but the trend for girls did not show significant differences during the same years [12].

Physical activity may have an effect on life satisfaction, self-rated health and peer support. The HBSC surveys collect data on these aspects of adolescents’ mental wellbeing and the comparison between Italian and international HBSC 2018 data showed similar prevalence in mean life satisfaction among young people (Italian, 7.6 and international, 7.8, presented in the international HBSC Report as the average score on a scale of 0–10) [22]. Boys and younger adolescents from the international HBSC 2018 reported higher levels of life satisfaction, and these findings were similar to those reported by Italian adolescents.

Similarly, the results on self-rated health showed a comparable prevalence of Italian and adolescents from other countries who reported excellent health perception (Italian, 35% and international, 37%). Both Italian and international data showed a decline from 11- to 15-year-old adolescents; girls tended to claim a lower level of excellent health perception than boys and almost all countries/regions participating in the HBSC 2018 survey found similar results [22].

The results showed a pronounced gender gap in relation to health outcomes, such as life satisfaction and self-rated health among adolescents. Previous scientific reports also pointed out comparable results on differences between boys and girls [25,26,27,38,39], and the data evaluation highlighted the possible implication of both physical and social differences in life satisfaction and health perception between girls and boys, during early adolescence [40]. In particular, differences in the timing of pubertal maturation, the resulting physical changes and their perception might contribute to a low level of life satisfaction and health perception among girls and, consequently, determine age differences as well. Social and cultural environment might also play a role in this gender gap; major control from parents and a higher level of school pressure could be associated with lower levels of psychological wellbeing among girls.

Regarding social wellbeing, the findings on peer support revealed a higher prevalence of high peer support among Italian adolescents, as compared with those from the international HBSC 2018 (Italian, 66% and international, 60%, respectively). Overall, both Italian and international data highlighted that girls were more likely to claim high levels of peer support in all age groups. A decrease from 11 to 13 years old was observed among Italian girls (Italian, 74% vs. international, 60%) and, less noticeably, among boys from other countries (Italian, 57% vs. international, 54%) [22].

In accordance with other studies, our results underlined that a high level of physical activity was associated with better psychological and social wellbeing in adolescents [14,23,24]. Our results showed a more marked effect of VPA on high levels of life satisfaction, good/excellent perception of health and peer support than MVPA for girls and boys. The data on the positive association between the level of VPA and life satisfaction were in agreement with those recently reported for adolescents from other countries involved in the HBSC survey [26,27]. Our findings showed that the effect of VPA on high levels of life satisfaction was more marked among boys and differences related to age were found between boys and girls. In detail, the results about VPA and its major influence on boys were in agreement with those reported for adolescents aged 11–13 years in Lithuania [25]. The impact of physical activity on life satisfaction increased with age among boys. In contrast, girls showed a lower association between VPA and life satisfaction from 11 to 15 years. The results for adolescents aged 10–17 years in Ireland showed that VPA was stronger in girls than boys [26]. Given that girls tended to claim a lower level of life satisfaction and, in general, a lower level of psychological health, the authors emphasized the need to encourage physical activity, especially among girls. The results on Italian adolescents were in agreement with those reported by Molcho et al. (2021), confirming the importance of promoting a more active lifestyle, in particular, VPA among girls [26].

The results on adolescents’ health perception showed a strong association with VPA. The same trend was observed for boys and girls, indicating a more marked influence of vigorous physical activity among older boy and girls. The importance of engagement in sport among adolescents in association with high levels of health perception was also reported by a recent investigation on Slovak adolescents. The authors highlighted a more pronounced association between VPA and good/excellent self-rated health among adolescents who participated in vigorous physical activity several times per week (four–six times a week and every day) [27].

Our results found that physical activity was positively associated with peer support. Involvement in both MVPA and VPA had a positive impact on social wellbeing in adolescents. In contrast to life satisfaction and health perception, no differences were found between MVPA and VPA, indicating that both physical activities may improve the relationships among adolescents and promote their cohesion. Our results are in agreement with those previously reported in the literature [41,42]. During adolescence, peer relationships increase and become more intense; young people who perceive supportive friendships may achieve a better psychological wellbeing and, at the same time, reduce the negative effects of stressful situations [43,44,45]. Physical activity might play an important role in promoting adolescents’ interactions and supportive friendships.

As with physical activity, socioeconomic status may have an impact on psychological and social wellbeing in young people. Adolescents from high-affluence families tend to experience major levels of life satisfaction, health perception and support from friends. According to our results, data from the international HBSC 2018 showed differences between adolescents’ life satisfaction and self-rated health, in relation to FAS [22]. With the exception of some countries, adolescents form more affluent families reported a higher level of life satisfaction, as well as better health. The association between FAS and mental wellbeing was found to be more pronounced for life satisfaction. In the international HBSC 2018, the same association between socioeconomic status of families and peer support was revealed; in more than half of the countries, high FAS was positively associated with better peer relationships and, in some countries, this association was found highly noticeable. Moreover, socioeconomic status has been previously reported as a factor that might modulate the physically active lifestyle of adolescents [41,46,47,48,49,50]. The international HBSC 2018 data showed that socioeconomic status may have an impact on physical activity levels among adolescents. In a large majority of countries, boys and girls from higher-affluence families reported higher levels of MVPA and VPA [22]. Overall, these findings underlined differences in adolescents’ psychological and social wellbeing, related to their socioeconomic condition, suggesting the importance of considering this aspect in prevention and intervention programmes, focusing on the enhancement of psychosocial wellbeing, for instance, through physical activity.

## 5. Conclusions

The results of our study highlighted the importance of improving participation in moderate-to-vigorous and vigorous physical activity during adolescence. In particular, focused strategies and actions should take into consideration age, gender and socioeconomic differences. Older adolescents and girls who claimed lower levels of MPVA and VPA need particular attention. Stakeholders and policymakers should enhance the knowledge of the benefits of a more active lifestyle, as well as implementing policies at the national and local level. Concerning girls, their lower levels of life satisfaction and health perception and the improvement associated with greater physical activity should be taken into account when addressing policies and focused actions to promote healthier lifestyles.

Specific attention should be paid to inequalities in MVPA and VPA related to socioeconomic status. The challenge is to introduce policies at school and community level, able to involve adolescents from less affluent families in physical activities also. In particular, the opportunities for physical activity should be improved by school curricular and school-based extracurricular activities, as well as community initiatives, also accessible to families with a low socioeconomic level.

### Strengths and Limitations

The main strengths are standardized and validated data collection procedures, based on the international HBSC study, and the use of a large and representative Italian sample to investigate the level of physical activity among Italian adolescents aged 11, 13 and 15, as well as the association between physical activity and psychosocial wellbeing, with a low percentage of missing values.

Limitations of the HBSC are the cross-sectional design, which does not allow one to draw conclusions about causation, and the self-reported information.

## Figures and Tables

**Table 1 ijerph-19-04799-t001:** MVPA, VPA, life satisfaction, self-rated health, peer support and FAS of the sample by age and gender. Italy, 2018.

	11 Years Old *n*= 19,504		13 Years Old *n* = 20,554		15 Years Old *n* = 18,918		All Age Groups *n* = 58,976	
	Boys (%)	Girls (%)	Boys (%)	Girls (%)	Boys (%)	Girls (%)	Boys (%)	Girls (%)
**MVPA during last seven days**								
Never	4.5	5.5	6.0	12.1	10.0	17.9	6.5	11.4
One day	5.5	7.6	5.6	9.5	8.8	13.0	6.4	9.8
Two days	16.9	23.5	16.0	22.6	15.8	19.9	16.3	22.2
Three days	18.7	20.3	19.0	20.1	20.6	19.8	19.3	20.1
Four days	17.6	17.0	19.0	15.0	16.9	12.6	17.9	15.0
Five days	13.6	11.2	13.9	9.8	12.4	7.7	13.4	9.7
Six days	8.3	6.0	8.0	4.9	7.1	4.0	7.9	5.1
Seven days	14.8	8.9	12.5	6.0	8.5	5.1	12.2	6.8
Missing	1.7	1.0	1.1	0.6	0.6	0.4	1.1	0.7
**MVPA during last seven days**								
MVPA at least four days a week	54.3	43.1	53.4	35.7	44.8	29.4	51.3	36.5
MVPA less than four days a week	47.3	56.9	46.6	64.3	55.2	70.6	48.7	63.5
**VPA in free time, outside school hours**								
Once a month/Less than once a month/Never	12.6	16.3	13.7	23.9	16.9	28.9	14.1	22.6
Once a week	10.2	15.6	11.4	15.9	13.3	16.0	11.5	15.8
2 to 3 times a week	37.1	42.4	37.8	39.5	37.6	37.3	37.5	39.9
4 to 6 times a week	23.7	15.6	24.5	14.4	23.1	13.2	23.8	14.5
Every day	16.4	10.1	12.6	6.3	9.1	4.6	13.1	7.2
Missing	1.0	0.5	0.6	0.3	0.6	0.3	0.7	0.4
**VPA in free time, outside school hours**								
VPA at least 2 days a week	77.2	68.1	74.9	60.2	69.8	55.1	74.4	61.6
VPA less than 2 days a week	22.8	31.9	25.1	39.8	30.2	44.9	25.6	38.4
**Life satisfaction (Cantril ladder): greater than or equal to six**								
Yes	91.5	89.0	91.1	85.5	88.2	82.2	90.4	85.8
No	8.5	11.0	8.9	14.5	11.8	17.8	9.6	14.2
Missing	1.0	1.0	0.7	0.6	1.1	0.5	0.9	0.7
**Self-rated health: good/excellent**								
Yes	92.8	93.3	92.8	89.5	90.0	84.3	92.0	89.4
No	7.2	6.7	7.2	10.5	10.0	15.7	8.0	10.6
Missing	0.9	0.5	0.5	0.2	0.7	0.1	0.7	0.3
**Peer support: Friends try to help, can count on friends, share joys and sorrows with friends, talk about problems with friends. High peer support (≥5.5)**								
Yes	62.5	74.1	60.9	70.1	60.2	70.2	61.3	71.6
No	37.5	25.9	39.1	29.9	39.8	29.8	38.7	28.4
Missing	2.7	2.2	2.6	1.1	1.8	0.9	2.4	1.4
**Family Affluence Scale (FAS)**								
Low	28.0	29.7	27.5	30.1	29.8	31.3	28.3	30.3
Medium	47.0	47.8	46.8	47.1	47.9	47.9	47.2	47.6
High	25.0	22.5	25.7	22.8	22.3	20.8	24.5	22.1
Missing	3.5	2.7	3.6	1.9	2.8	1.6	3.3	2.1

**Table 2 ijerph-19-04799-t002:** Logistic regression models for life satisfaction, self-rated health and peer support, stratified for gender and age. OR (IC95%).

Boys				Girls		
	11 Years Old	13 Years Old	15 Years Old	11 Years Old	13 Years Old	15 Years Old
**Life Satisfaction ≥ 6**						
**MPVA less than four days a week**	1	1	1	1	1	1
**MPVA at least four days a week**	1.34 (0.99–1.83)	**1.59** **(1.22–2.07)**	1.16 (0.92–1.46)	0.87 (0.66–1.15)	1.23 (0.96–1.58)	1.25 (0.99–1.57)
**VPA Less than 2 days a week**	1	1	1	1	1	1
**VPA at least 2 days a week**	1.29 (0.96–1.73)	**1.38** **(1.05–1.82)**	**1.64** **(1.29–2.10)**	**1.36** **(1.05–1.76)**	**1.27** **(1.03–1.57)**	1.19 (0.98–1.43)
**FAS: low**	1	1	1	1	1	1
**FAS: medium**	1.07 (0.80–1.43)	**1.62** **(1.24–2.13)**	**1.39** **(1.12–1.72)**	**1.62** **(1.20–2.18)**	1.26 (0.99–1.60)	**1.59** **(1.32–1.93)**
**FAS: high**	1.07 (0.74–1.54)	**1.86** **(1.31–2.65)**	**1.93** **(1.42–2.63)**	**1.72** **(1.24–2.38)**	**1.80** **(1.35–2.40)**	**1.45** **(1.14–1.86)**
**Self-rated health: good/excellent**						
**MPVA less than four days a week**	1	1	1	1	1	1
**MPVA at least four days a week**	**2.23** **(1.62–3.07)**	**1.56** **(1.20–2.03)**	**1.57** **(1.21–2.05)**	1.20 (0.89–1.62)	**1.53** **(1.14–2.06)**	**1.37** **(1.09–1.71)**
**VPA less than 2 days a week**	1	1	1	1	1	1
**VPA at least 2 days a week**	1.31 (0.98–1.74)	**1.75** **(1.30–2.34)**	**2.48** **(1.92–3.21)**	**1.48** **(1.11–1.97)**	**1.48** **(1.15–1.91)**	**1.67** **(1.37–2.03)**
**FAS: low**	1	1	1	1	1	1
**FAS: medium**	1.21 (0.88–1.67)	**1.52** **(1.13–2.03)**	1.10 (0.86–1.40)	**1.49** **(1.10–2.03)**	1.26 (0.99–1.60)	1.20 (0.97–1.47)
**FAS: high**	1.15 (0.76–1.75)	**1.43** **(1.00–2.04)**	**1.63** **(1.17–2.28)**	1.27 (0.89–1.82)	1.34 (0.98–1.82)	**1.42** **(1.10–1.83)**
**Peer support ≥ 5.5**						
**MPVA less than four days a week**	1	1	1	1	1	1
**MPVA at least four days a week**	**1.31** **(1.12–1.54)**	**1.52** **(1.31–1.77)**	**1.22** **(1.07–1.40)**	**1.21** **(1.00–1.46)**	1.07 (0.89–1.27)	**1.23** **(1.02–1.47)**
**VPA in free time, outside school hours: Less than 2 days a week**	1	1	1	1	1	1
**VPA in free time, outside school hours: at least 2 days a week**	**1.33** **(1.12–1.58)**	**1.37** **(1.15–1.64)**	**1.47** **(1.26–1.71)**	**1.37** **(1.13–1.66)**	**1.32** **(1.13–1.55)**	1.03 (0.89–1.19)
**FAS: low**	1	1	1	1	1	1
**FAS: medium**	**1.22** **(1.02–1.46)**	1.07 (0.91–1.24)	0.93 (0.79–1.09)	1.11 (0.91–1.34)	**1.32** **(1.11–1.57)**	1.08 (0.92–1.26)
**FAS: high**	**1.31** **(1.05–1.63)**	**1.39** **(1.14–1.71)**	1.13 (0.93–1.38)	**1.42** **(1.09–1.85)**	**1.53** **(1.23–1.90)**	**1.26** **(1.02–1.55**)

Statistically significant results are in bold.

## Data Availability

The data presented in this study are available in accordance with the Italian HBSC data access policy. Requests should be directed to paola.nardone@iss.it, member of the National Centre for Disease Prevention and Health Promotion, Italian National Institute of Health.

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
