# Peer review of "Physical Activity among Italian Adolescents: Association with Life Satisfaction, Self-Rated Health and Peer Relationships"

_ijerph, 2022, doi:10.3390/ijerph19084799_

Round 1

Reviewer 1 Report

Authors present a manuscript adding current and well explained descriptive results from the association of physical activity, life satisfaction and self-rated health in the Italian context. The results indicate that relations are in the same verge that other international studies and it, generally, well substantiated. In this sense, it could be a good contribution for the research field that might be disseminated.  

Nonetheless, there are some suggestions and comments that authors should attend to provide a definitive consideration of the publication.

Introduction:

Bearing in mind that authors take as a reference the 2020 WHO recommendations, some references to sedentary time recommendations should be added, even if sedentary behaviour was not one of the study variables. Moreover, the WHO recommendations are contemporary of other relevant recommendations such as the Canadian Society one’s. These guidelines, add the convenience of assessing the daily behaviour among the 24 hours, paying also attention to the sleep time. Therefore, this is another important element to incorporate to the introduction, together with some references towards the relation among physical activity and socioeconomic status (in both abstract and introduction section), due to is another relevant sociodemographic variable attended in the study.

Related with this last suggestion, authors state at the aims of the study that they explore peer relationships of the main variables (Physical activity, life satisfaction and self-rated health), but I suggest they must include explicitly that comparisons will hold by gender and socioeconomic status.

Materials and method:

The section is well explained and detailed but I suggest that it is convenient indicate when the Odds-Ratio differences are statistically significant regarding the values of the confidence intervals. In fact, these significant differences should be highlighted in the table of results.  

Conclusions:

As authors do at the discussion section, it is advisable to add some insights for policymakers to considerer the relevance and the equality in the access of physical activity, especially in deprived contexts.

Author Response

Response to Reviewer 1 Comments

Dear reviewer,

many thanks for your review of our manuscript. We revised the manuscript by modifying the sections based on your suggestions and comments. Our case-by-case responses to the reviewer suggestion and comments were also provided. Revisions in the text are highlighted in red. We hope that the revised manuscript will now be deemed acceptable for publication in the International Journal of Environmental Research and Public Health-Special Issue "Promoting Adolescent Health and Wellbeing for a Better Transition to Healthy Lifestyle Adulthood".

Point 1: Introduction: bearing in mind that authors take as a reference the 2020 WHO recommendations, some references to sedentary time recommendations should be added, even if sedentary behaviour was not one of the study variables. Moreover, the WHO recommendations are contemporary of other relevant recommendations such as the Canadian Society one’s. These guidelines, add the convenience of assessing the daily behaviour among the 24 hours, paying also attention to the sleep time.

Response 1: We would like to thank you for your constructive suggestion. We added the reccommendations of the Canadian Society for Exercise Physiology and the new reference [9] in the introduction section. Please check the line 52-56.

Point 2: introduction: Therefore, this is another important element to incorporate to the introduction, together with some references towards the relation among physical activity and socioeconomic status (in both abstract and introduction section), due to is another relevant sociodemographic variable attended in the study.

Response 2: we really appreciate your helpful comment. We added in introduction section the relation among physical activity and socioeconomic status. We also added the reference related to the international HBSC 2018 report [reference 18]. We added in both abstract and introduction section the association of socioeconomic status (FAS) with psychological and social wellbeing of adolescents, as studied by logistic regression reported in this manuscript. Please check the lines 104-110. Other references were reported in the discussion section (line 357-376) in which we explained this previously mentioned association and the importance to take into account socioeconomic status due to its impact on psychological and social wellbeing as well as physical activity levels of adolescents (references 38, 43-47).

Point 3: introduction: Related with this last suggestion, authors state at the aims of the study that they explore peer relationships of the main variables (Physical activity, life satisfaction and self-rated health), but I suggest they must include explicitly that comparisons will hold by gender and socioeconomic status.

Response 3: Thank you for your suggestion. We added in aim of the study the stratification by gender and age. We also included in the aim of the paper the study of association between FAS and self-rated health, life satisfaction and peer relationships (line 116-120).

Point 4: materials and method: the section is well explained and detailed but I suggest that it is convenient indicate when the Odds-Ratio differences are statistically significant regarding the values of the confidence intervals. In fact, these significant differences should be highlighted in the table of results.

Response 4: thank you for your suggestion. We highlighted in the Table 2 statistically significant results.

Point 5: conclusions: as authors do at the discussion section, it is advisable to add some insights for policymakers to considerer the relevance and the equality in the access of physical activity, especially in deprived contexts.

Response 5: we really appreciate your helpful suggestion for manuscript conclusions. We added some insights for policymakers in relation to the importance to create opportunities in physical activities also for adolescents from less affluent families (lines 387-392).

Reviewer 2 Report

Review: Physical activity among Italian adolescents: association with life sat-isfaction, self-rated health and peer relationships. 

At the beginning of the review, I would like to thank the author(-s) for their interesting manuscript. I found the submitted paper to be worthy of interest for researchers, coaches and persons from anti-doping organizations. I’ve red it carefully and in general I think that it adds some new insights about PA of adolescents.

Appreciating the scientific value of the article, let me make some comments and suggestions:

Line 64-66 information about the impact of PA on particular areas of activity / health dimensions is already given earlier. Although, I believe, the authors wanted to refer specifically to the period of adolescence, however reading this part of the text one gets the impression of a kind of redundancy, which it would be good to correct, giving the narrative a more coherent structure

Line 67-70 This seemingly obvious thesis, however, is being called into question by many studies which even show that time spent watching TV and being physically active are not related. Hence the postulates not to oppose passive and active behaviors as alternatives and to treat them as ends of the same continuum of energy expenditure. These may be two independent phenomena, the more so that in some experiments on the effectiveness of TV addiction reduction programs, the success in this area was not accompanied by an increase in the level of physical activity. I would therefore recommend mentioning this controversy as being more "honest" to its readers

Line 82 once again the remark that the same effects had happened before

Line 157-160 this type of information belongs to the Material and methods section, as it is a description of the surveyed population and not results per se

Note to the description of tables: the symbol of the capital letter N applies to the entire study population, the symbol of the lower case letter "n" is used for the size of the subgroups

The authors only mention one limitation of their research. And is it not the time of their conduct, especially in the face of the SARS-CoV-2 pandemic, which brought about big changes in human activity?

The discussion of the results is correct, because the authors present their results in comparison with other studies conducted in the context of other studies conducted within the HBSC project. While you can have reservations about the "construction" of some research tools, it does not change the fact that they are used throughout the project.

Author Response

Response to Reviewer 2 Comments

Dear reviewer,

many thanks for your review of our manuscript. We revised the manuscript by modifying the sections based on your suggestions and comments. Our case-by-case responses to the reviewer suggestion and comments were also provided. Revisions in the text are highlighted in red. We hope that the revised manuscript will now be deemed acceptable for publication in the International Journal of Environmental Research and Public Health-Special Issue "Promoting Adolescent Health and Wellbeing for a Better Transition to Healthy Lifestyle Adulthood".

Point 1: line 64-66 information about the impact of PA on particular areas of activity / health dimensions is already given earlier. Although, I believe, the authors wanted to refer specifically to the period of adolescence, however reading this part of the text one gets the impression of a kind of redundancy, which it would be good to correct, giving the narrative a more coherent structure

Response 1: we would like to thank you for your constructive suggestion. We changed the text as suggested (line 67-73).

Point 2: line 67-70 this seemingly obvious thesis, however, is being called into question by many studies which even show that time spent watching TV and being physically active are not related. Hence the postulates not to oppose passive and active behaviours as alternatives and to treat them as ends of the same continuum of energy expenditure. These may be two independent phenomena, the more so that in some experiments on the effectiveness of TV addiction reduction programs, the success in this area was not accompanied by an increase in the level of physical activity. I would therefore recommend mentioning this controversy as being more "honest" to its readers

Response 2: we really appreciate your helpful suggestion. We added in the introduction section the text in relation to the reviewer suggestion, as well as the new references [18, 19]. Please check the line 78-79.

Point 3: line 82 once again the remark that the same effects had happened before

Response 3: thank you for your saggestion. We like remarking this issue which introduce the main aim of the paper, that is, the possible impact of physical activity on some mental health outcomes of adolescents.

Point 4: line 157-160 this type of information belongs to the Material and methods section, as it is a description of the surveyed population and not results per se

Response 4: thank you for your comment. We consider the sample size data as the first output for the subsequent analysis. For this reason, we also included in line 178-180 the description of Table 1 which shows the characteristics of the sample by age and gender.

Point 5: note to the description of tables: the symbol of the capital letter N applies to the entire study population, the symbol of the lower case letter "n" is used for the size of the subgroups

Response 5: we really appreciate your helpful comment. We included in Table 1 “n” in place of “N”.

Point 6: the authors only mention one limitation of their research. And is it not the time of their conduct, especially in the face of the SARS-CoV-2 pandemic, which brought about big changes in human activity?

Response 6: thank you for your suggestion. Data were collected from the 2018 Italian HBSC survey, before SARS-CoV-2 pandemic. By the next 2022 HBSC wave a comparison will be carried out taking into consideration changes due to pandemic period. Moreover, the next 2022 HBSC wave will also include questions about SARS-CoV-2 pandemic to explore the possible changes in life of adolescents and their families.

Point 7: the discussion of the results is correct, because the authors present their results in comparison with other studies conducted in the context of other studies conducted within the HBSC project. While you can have reservations about the "construction" of some research tools, it does not change the fact that they are used throughout the project.

Response 7: we really appreciate your helpful comment for manuscript discussion. The comparison of the results was performed especially with other studies from international 2018 HBSC network and, in particular, with international 2018 HBSC report which includes data from 50 countries across Europe and North America, all adhering to a detailed international study protocol.